# OpenReview forum: "Pruning for Protection: Increasing Jailbreak Resistance in Aligned LLMs Without Fine-Tuning"
_EMNLP/2024/Workshop/BlackBoxNLP — BlackboxNLP 2024_

### Official Review · Reviewer_W6NS · 2024-08-26

**Overall Assessment:** 4
**Confidence:** 3

**Best Paper:**

2

**Best Paper Justification:**

Safety is a critical concern for LLMs, and the proposed evaluation provides evidence that an existing method may exist for efficiently aligning an LLM for safety.

**Comments Questions Suggestions And Typos:**

* How efficient is the WANDA method for pruning some percentage of the LLM's parameters? Do the authors have any empirical runtime results?

**Paper Summary:**

The authors propose using WANDA, a technique for pruning parameters, to prune up to 30% of an LLM's parameters to reduce successful jailbreaking attempts. The experimental results suggest that pruning up to 30% of an LLM's parameters increases the attention interaction between malicious tokens and increases the perplexity for jailbreak prompts, and results in fewer successful jailbreak prompts.

**Summary Of Strengths:**

* The paper tackles an important problem of LLM safety alignment by proposing a method to reduce the number of successful jailbreaking prompts.
* The main experimental results performed on a wide range of malicious tasks using three different 7B parameter LLMs suggest pruning reduces successful jailbreak attempts while maintaining comparable predictive performance to a model with no pruning.
* It's interesting to see that pruning starts to have an adverse effect after pruning ~30%+ parameters.
* The authors perform further analyses that suggests pruning sharpens attention mechanisms on malicious tokens and increases perplexity on jailbreak prompts only.

**Summary Of Weaknesses:**

* I appreciate the in-depth analysis of the results, however, it's still not clear to me how increased attention between malicious tokens and increased perplexity reduces the number of successful jailbreak attempts? That is, how does increased attention between malicious tokens lead to a generated text that refuses to respond to the request?
* It's not entirely clear to me how WANDA prunes the LLM's parameters, perhaps the authors can provide a brief description of WANDA as that is the main method used to prune the LLMs in the experiments.

---

### Official Review · Reviewer_SiuR · 2024-09-09

**Overall Assessment:** 3
**Confidence:** 3

**Best Paper:**

1

**Best Paper Justification:**

N/A

**Comments Questions Suggestions And Typos:**

- For Figure 1, I suggest that the authors position it at the top of the page.

**Paper Summary:**

This paper investigates the problems of how model compression (specifically the WANDA method) influences the performance of defenses against jailbreaking. Results show that moderate pruning (approximately 20%) showcases an increase in jailbreaking performance, which shares similar trends with the performance line. The authors argue that this phenomenon is correlated with the sharp attention pattern across the layer.

**Summary Of Strengths:**

- I believe this is an interesting research topic that deserve much attention.
- The results are somewhat promising

**Summary Of Weaknesses:**

- The biggest comment for me is that the writing has many spaces to improve. For example, from lines 501 to 512, the formalism is kind of unorganized and lacks some degree of rigor.
- I am curious how correlated the interpretation methods you used are with the actual experimental results you observed. I think this correlation is specific to the design choice of WANDA and the properties of the transformer it applied to. More theoretical analysis would be appreciated.

---

### Official Review · Reviewer_G6j1 · 2024-09-10

**Overall Assessment:** 4
**Confidence:** 4

**Best Paper:**

1

**Best Paper Justification:**

N/A

**Comments Questions Suggestions And Typos:**

Why do the static jailbreaking tests and GCG seem to show opposite trends?

**Paper Summary:**

This paper shows that Llama-2 Chat and Vicuna pruned via WANDA are more robust to jailbreaking attacks than unpruned models at low pruning rates. Meanwhile, pruning does not hurt model capabilities, as measured through standard benchmarks. Overall, this demonstrates an unexpected advantage to pruning, which is usually motivated by efficiency considerations. Finally, the paper analyzes the attention patterns of Llama-2, and shows that the robustness to jailbreaks correlates with a tendency of attention to ignore malicious tokens in the prompt.

One weakness is that the results sometimes don't follow a very clean pattern. Mistral does not benefit from pruning, unlike Llama-2 and Vicuna. There are also some somewhat contradictory findings, as greedy search-based jailbreaking approaches succeed against models pruned with low pruning rates, but fail when a higher pruning rate is used--the opposite of the static jailbreaking approaches.

I think these findings are useful and would be a very nice addition to the workshop. I can see how these results could be hard to build a complete narrative around for a main conference paper, but I support this as a workshop paper.

**Summary Of Strengths:**

* Surprising findings about robustness after pruning for two LLMs
* In comparison, benchmark scores do not change, so this cannot be explained by the LMs failing to understand the prompt
* Interpretability study of attention sharpness

**Summary Of Weaknesses:**

* Findings hold for Llama-2 and Vicuna but not Mistral. Authors hypothesize that models with more safety tuning are more likely to exhibit benefits after pruning, but this could also be specific to the Llama family, as Vicuna is fine-tuned from Llama.
* GCG results seem to contradict the main story--it finds that robustness only kicks in at very high pruning rates.
* Section 7 (OLS analysis) seems very different, as it studies pruning of a very different type of model. I'm not convinced the analysis carries over.

---

### Decision · Program_Chairs · 2024-09-20

**Decision:**

Accept

**Comment:**

The paper tackles an important topic and the reviewers are all in agreement that the paper presents a valuable addition to the BlackboxNLP workshop.